# Lack of PCSK6 Increases Flow-Mediated Outward Arterial Remodeling in Mice

**DOI:** 10.3390/cells9041009

**Published:** 2020-04-18

**Authors:** Samuel Röhl, Bianca E. Suur, Mariette Lengquist, Till Seime, Kenneth Caidahl, Ulf Hedin, Anders Arner, Ljubica Matic, Anton Razuvaev

**Affiliations:** 1Department of Molecular Medicine and Surgery, Karolinska Institutet, 171 76 Stockholm, Sweden; Samuel.Rohl@ki.se (S.R.); bianca.suur@ki.se (B.E.S.); mariette.lengquist@ki.se (M.L.); till.seime@ki.se (T.S.); kenneth.caidahl@ki.se (K.C.); ulf.hedin@ki.se (U.H.); 2Institute of Medicine, Sahlgrenska Academy, University of Gothenburg, 413 45 Gothenburg, Sweden; 3Department of Clinical Sciences Lund, Thoracic Surgery, Lund University, 221 84 Lund, Sweden; anders.arner@med.lu.se

**Keywords:** vessel wall remodeling, pro-protein convertase subtilisin/kexin 6, smooth muscle cells

## Abstract

Proprotein convertases (PCSKs) process matrix metalloproteases and cytokines, but their function in the vasculature is largely unknown. Previously, we demonstrated upregulation of PCSK6 in atherosclerotic plaques from symptomatic patients, localization to smooth muscle cells (SMCs) in the fibrous cap and positive correlations with inflammation, extracellular matrix remodeling and cytokines. Here, we hypothesize that PCSK6 could be involved in flow-mediated vascular remodeling and aim to evaluate its role in the physiology of this process using knockout mice. Pcsk6^−/−^ and wild type mice were randomized into control and increased blood flow groups and induced in the right common carotid artery (CCA) by ligation of the left CCA. The animals underwent repeated ultrasound biomicroscopy (UBM) examinations followed by euthanization with subsequent evaluation using wire myography, transmission electron microscopy or histology. The Pcsk6^−/−^ mice displayed a flow-mediated increase in lumen circumference over time, assessed with UBM. Wire myography revealed differences in the flow-mediated remodeling response detected as an increase in lumen circumference at optimal stretch with concomitant reduction in active tension. Furthermore, a flow-mediated reduction in expression of SMC contractile markers SMA, MYH11 and LMOD1 was seen in the Pcsk6^−/−^ media. Absence of PCSK6 increases outward remodeling and reduces medial contractility in response to increased blood flow.

## 1. Introduction

Vascular remodeling is a fundamental biological process for maintaining vessel wall homeostasis in response to alterations in blood flow, inflammation or injury [1,2]. In atherosclerosis, chronic inflammation in the affected vessel wall results in constant tissue remodeling [3]. Alterations in local inflammation or blood flow may also induce an increased extracellular matrix (ECM) modulation in the shoulder region of the fibrous cap, resulting in plaque destabilization with subsequent risk of rupture [2,3,4]. Furthermore, improper vascular remodeling is detrimental for long-term patency following surgical interventions [5,6]. Despite differences in hemodynamics, anatomical location and surgical techniques, endovascular treatments share a similar panorama of complications related to excessive, inadequate or constrictive vascular remodeling [5,7]. Expansive, or outward, remodeling may result in aneurysm formation or even congestive heart failure in patients with arteriovenous dialysis fistulas, due to an excessive shunting of blood through the dialysis fistula [8,9]. Inadequate and constrictive, or inward, remodeling can result in insufficient blood flow for hemodialysis or vein graft failure with subsequent critical limb ischemia [5,7]. Hence, a pathological remodeling process increases the risk of treatment failure resulting in patient suffering, prolonged hospital stays and need for surgical re-intervention.

Vascular remodeling consists of a rearrangement of the ECM in order to uphold the vessel wall homeostasis in response to alterations in the chemical and biomechanical milieu [1,2]. Activated matrix metalloproteases (MMPs) degrade the ECM which results in structural disassembly, release of ECM-bound growth factors, such as transforming growth factor beta-1 (TGFB1) and platelet derived growth factor beta (PDGFB), and facilitate activation and migration of residing medial smooth muscle cells (SMCs) [10,11]. Activation of the TGFB1 and PDGFB signaling pathways stimulates a fibroproliferative response characterized by reduced collagen degradation and secretion of ECM components [11,12,13,14]. The activity of MMPs is regulated by tissue inhibitors of metalloproteinases (TIMPs) [2,10]. Hence, it has been speculated that pathological vascular remodeling is related to shifts in the MMP/TIMP ratio, in which an increased ratio causes expansive remodeling while decreased ratio results in constrictive remodeling [15,16]. Despite promising results in experimental animal studies, pharmacological modification of the MMP/TIMP ratio has not yet proven to be of clinical value [17].

Pro-protein convertase subtilisin/kexin 6 (PCSK6), also known as PACE4, is a serine protease which processes and activates target proteins through selective cleavage of their biologically inactive precursors [18,19]. The function of PCSK6 has been investigated in cancer and has shown to be associated with enhanced tumor invasiveness, cytokine release and MMP activation [20,21]. Previous studies have also reported that PCSK6 is of importance for the processing and activation of growth factors, primarily TGFB1 and PDGFB [22,23,24,25]. The influence of PCSK6 on the cardiovascular system and disease development still remains largely unknown, although variants in the PCSK6 genomic locus have been associated with congenital heart disease and aortic dissection [26,27] and PCSK6 has been implicated as a regulator of blood pressure in mice subjected to a sodium chloride enriched diet [28]. Utilizing a large human Biobank of Karolinska Endarterectomies, we have previously shown that upregulation of PCSK6 is associated with atherosclerotic plaque instability in patients with carotid stenosis [29]. More recently, we demonstrated that PCSK6 is a key factor in vascular remodeling through regulation of SMC migration and intimal invasion by modulation of MMP14 activation, particularly upon PDGF stimulation [25].

Here, we sought to further characterize the role of PCSK6 in vascular responses from a physiology perspective, using experimental vascular research models applied on Pcsk6 full knock-out mice [22]. In comparison to wild type controls, Pcsk6^−/−^ mice have been shown to have a slight increase in habitual systolic blood pressure without alterations in myocardial thickness or function [28]. Interestingly, deletion of Pcsk6 does not cause any obvious vascular phenotype detectable in adult Pcsk6^−/−^ mice despite some alterations in the overall gene expression, which indicates that compensatory mechanisms are present. However, we have previously shown that these compensatory mechanisms are insufficient to uphold the normal function of medial SMCs in response to alterations in the physiological milieu [25]. Over the past decades, several experimental models of flow-mediated vascular remodeling have been developed. The murine carotid artery ligation model was originally developed by A. Kumar et al. [30] in order to study the effect of low shear stress on intimal hyperplasia formation and constrictive remodeling in the ligated artery. However, it is also known that unilateral carotid ligation induces a 40–70% increase in blood flow to the contralateral un-ligated artery. This elevated blood flow generates an increase in shear stress on the arterial wall which induces a non-inflammatory flow-mediated remodeling response [31,32]. Utilizing modern non-invasive in vivo imaging techniques, such as ultrasound biomicroscopy (UBM), and ex vivo physiological measurements in wire myography, it is now possible to visualize morphological and hemodynamic changes over time and their relation to the physiological properties in a remodeled artery [33,34]. With this approach, here we aim to investigate the impact of PCSK6 deletion on arterial morphology, hemodynamics and physiology in the flow-mediated outward remodeling carotid ligation model.

## 2. Materials and Methods

### 2.1. Animals

Pcsk6^−/−^ mice were generously provided by Rachel E. Miller and Anne-Marie Malfait (Rush University Medical Center, Chicago, IL, US). These mice were generated as previously described and backcrossed onto C57BL/6J background for at least ten generations [22,35]. A total of 46 male animals with an age of 5–6 months were used in this study. The wild type (WT) mice (C57Bl/6J, *n* = 22) were purchased from Charles River (Scanbur Research A/S, Sollentuna, Sweden). The Pcsk6^−/−^ knockout (KO) mice (*n* = 24), after import from the USA, were bred at Karolinska Institute according to local protocols and standards. Twenty-eight animals (WT *n* = 14, KO *n* = 14) underwent left common carotid artery (CCA) ligation, 18 animals (WT *n* = 8, KO *n* = 10) served as controls. Analgesics (Buprenorphine, 0.01 mg/kg, Temgesic^®^, RB Pharmaceuticals Ltd., Berkshire, Great Britain) were administered if signs of pain or discomfort were observed. All animals were housed in a dedicated animal facility according to the institutional guidelines for animal care and monitored by professional animal caretakers. All animal experiments and breeding were reviewed and approved by the Regional Ethical Review Board of North Stockholm (ethical permit numbers N67/14 and Dnr 2241-2019).

### 2.2. Study Design

Three separate experiments were conducted in order to investigate the impact of Pcsk6 on flow-mediated remodeling. The first experiment included 20 animals (*n* = 10 per strain) of which 12 (*n* = 6 per strain) were subjected to carotid ligation in order to increase the blood flow (IF) in the contralateral CCA, while 8 (*n* = 4 per strain) served as controls. All animals underwent UBM examination (prior to surgery, 3 and 5 weeks after surgery) followed by euthanization at 6 weeks and tissue harvest for ex vivo wire myography (detailed description below). The second experiment included 12 animals (*n* = 6 per strain) of which 8 (*n* = 4 per strain) were subjected to carotid ligation and 4 (*n* = 2 per strain) served as control. Six weeks after ligation the animals were euthanized and tissue was harvested for transmission electron microscopy evaluation. In the third experiment, 8 animals (*n* = 4 per strain) were subjected to carotid ligation and 6 animals (WT *n* = 2, KO *n* = 4) served as control. The animals were euthanized after 6 weeks and tissue was harvested for histological and immunohistochemical analysis.

### 2.3. Carotid Ligation

To increase the blood flow in the right CCA, a ligation of the left CCA was performed [30]. General anesthesia was obtained using isoflurane (IsoFlo^®^Vet, Abbott Laboratories Ltd., Berkshire, England), 4% upon induction and 1.5% for maintenance. Following induction, the animals were placed in a supine position on a heating pad and the hair on the ventral part of the neck was removed using commercial hair removal cream. An oblique incision was performed from the angle of the left mandible towards the hyoid bone. The left carotid bifurcation was exposed by blunt dissection and the CCA was ligated proximal to the carotid bifurcation using a monofilament polypropylene suture (Surgipro 8-0, Auto Suture Company, Norwalk, CT, USA). The wound was closed using interrupted resorbable sutures (Vicryl 5-0, Johnson-Johnson International, Diegem, Belgium). Postoperative analgesia was administered by subcutaneous injection upon surgery. Successful ligation of the left CCA was confirmed in all mice by PW Doppler examination and/or visual inspection upon perfusion. The same experienced operator (SR) performed all surgeries.

### 2.4. Ultrasound Biomicroscopy

For UBM, a Vevo2100 (Fujifilms Visualsonics Inc., Toronto, ON, Canada) system equipped with a 30–70 MHz transducer (MS700) was used. All animals were examined at three different time points (prior to surgery, 3 and 5 weeks after ligation). The examinations were performed under general anesthesia, as described above. Upon ultrasound examination, the animals were positioned on an integrated heating table which allowed for continuous ECG and respiratory rate registration. Cine loops and pulse-wave (PW) Doppler data of the CCAs were saved on an external hard drive and analyzed using offline software (Vevolab 1.7). All measurements were performed 1–2 mm proximal to the carotid bifurcation, in triplicates (three cardiac cycles) and according to the leading edge principle [36]. The inclination Doppler angle upon examination was < 60 degrees. Successful carotid ligation was confirmed through Doppler examination. Systolic (SD) and diastolic (DD) lumen diameters were identified through visual inspection of the ultrasound image at end-diastole and end-systole according to ECG. Elasticity in terms of strain was calculated as: 100 × ((SD-DD)/DD). PW Doppler data of the right CCA was analyzed to identify peak systolic velocity (PSV), end diastolic velocity (EDV), mean velocity (MV) and mean velocity time integral (VTI). Resistive index (RI) and Pulsatility index (PI) were calculated as: RI = (PSV-EDV)/PSV and PI = (PSV-EDV)/MV [37,38]. Volume flow rate (VFR) was calculated as: VFR (cm^3^/min) = (heart rate × lumen area (mm^2^) × VTI (mm))/1000 [38]. Wall shear stress (WSS) was calculated as: WSS = 4nQ/πr^3^, using the volumetric blood flow rate (Q) and an assumed blood viscosity constant (n) of 0.033 poise [39]. Arterial circumference was calculated with the assumption of a symmetric lumen, where circumference = π × lumen diameter, in order to compare the morphometry in UBM with myography. Ultrasound examinations, image analysis and calculations were performed by the same experienced operator (SR).

### 2.5. Myography

Wire myography was performed 6 weeks after carotid ligation, with the same cohort of 20 mice used in the UBM experiment, in order to evaluate the arterial wall physiology in the right CCA and thoracic aorta. The animals were euthanized under high dose isoflurane. Following euthanization, the arteries were harvested and placed in cold Krebs-Ringer solution (composition in mM: NaCl 123, KCl 4.7, KH_2_PO_4_ 1.2, MgCl_2_ 1.2, NaHCO_3_ 20, CaCl_2_ 2.5, glucose 5.5). The CCAs were cleaned from surrounding tissue under dissection microscope, the distal part of the CCAs was then divided forming 2 arterial rings with a length of 1.0–1.5 mm, which were mounted onto the myography jaws, using stainless steel wires (30 µm in diameter) on a myograph system (610M, DMT, Aarhus, Denmark). A similar preparation was performed for the thoracic aorta, where arterial ring segments with a length of 3–4 mm were mounted onto the myography jaws using pins. The myography wells were filled with 36.5 °C Krebs-Ringer solution and gas was constantly added for oxygenation and buffering (95%/5% O_2_/CO_2_, giving a pH of 7.4). The myograph was calibrated and the mounted vessels stretched to a passive tension, 2–3 mN (CCA) and 3–4 mN (thoracic aorta), and left for 45 min of equilibration. Following equilibration, the arteries were checked for viability through contraction for 5 min using addition of 80 mM KCl followed by 10 min relaxation through rinsing with Krebs-Ringer solution; this step was repeated 2–3 times until reproducible contractions were obtained. Thereafter, a length-force experiment was performed using repeated 3 min contractions with 80 mM KCl followed by rinsing and 7 min relaxation at increasing arterial circumference until the optimal circumference (optimal stretch) was identified, i.e., the circumference producing the maximal active wall tension (mN/mm). Once the optimal circumference was identified, the arteries were exposed to increasing concentrations of phenylephrine from 0.01–30 µM and contractile response was recorded. The arteries were then dilated using sodium nitroprusside 0.1 mM, rinsed in Krebs-Ringer solution and put in fixative (4% Zn-formaldehyde) at optimal circumference for 12 h and then dehydrated in 70% ethanol until embedment in paraffin blocks. Circumference was calculated as: (2 × *D*) + (2 × *w*) + (π × *w*), where *D* represents the distance between the myography jaws and *w* represents the wire or pin diameter. Active and passive tension (mN/mm) was calculated as: Force/(2 × arterial ring length) [40].

### 2.6. Transmission Electron Microscopy

A total of 12 mice (*n* = 6 per strain) were used, 8 mice (4 per strain) were subjected to carotid ligation and 4 mice (*n* = 2 per strain) served as control. Six weeks after surgery the animals were euthanized using high dose isoflurane. Upon euthanization the thoracic cage was opened and the pericardium dissected. The animals were perfused through left ventricular puncture, using the incised right auricle as outlet, at a pressure of 90 mmHg. Blood was removed through perfusion with saline for 3 min followed by 3 min perfusion with fixative (Glutaraldehyde 2.5% in 0.1 M Phosphate buffer, pH 7.4 at room temperature). The RCCA was then dissected, removed and stored in fixative at 4 °C. The specimens were rinsed in 0.1 M phosphate buffer, pH 7.4 and postfixed in 2% osmium tetroxide 0.1 M phosphate buffer, pH 7.4 at 4 °C for 2 h, dehydrated in ethanol followed by acetone and embedded in LX-112 (Ladd, Burlington, VT, USA). Semi-thin sections (0.5 µm) were cut and stained with toluidine blue O and used for light microscopic analysis. Ultrathin sections (approximately 50–60 nm) were cut by a Leica EM UC 6 (Leica, Wien, Austria) and contrasted with uranyl acetate followed by lead citrate and examined in a Hitachi HT 7700 (Tokyo, Japan) at 80 kV. Digital images were taken using a Veleta camera (Olympus Soft Imaging Solutions, GmbH, Münster, Germany). The images were analyzed with ImageJ analysis software (Wayne Rasband, National Institute of Mental Health, Bethesda, MD, USA). Analysis of the length of the elastic membrane per media area was performed according to the following formula: total elastic lamina length (µm)/total tunica media area (µm^2^).

### 2.7. Histochemical Staining

A total of 8 mice (*n* = 4 per strain) were subjected to carotid ligation and euthanized 6 weeks after surgery, and 6 animals (WT *n* = 2, KO *n* = 4) served as control. Upon euthanization, the thoracic cage was removed and the animal perfused with saline as described above. The right CCA was then dissected, removed and put in fixative for 24h (4% Zn-formaldehyde) followed by dehydration and paraffin-embedding. The arteries were then sectioned (5 μm), stained with Masson trichrome staining kit (Sigma-Aldrich) and Weigert-Van Gieson staining (Sigma-Aldrich) according to manufacturer directions and local protocol. The slides were examined using a Nikon Eclipse E800 microscope (Nikon Instruments Europe BV, Amsterdam, Netherlands) and images captured using an INFINITY3-6URC digital camera and INFINITY ANALYZE 7 software (Teledyne Lumenera, Ottawa, Canada). Image analysis was performed using ImageJ analysis software. 

For immunostainings, all immunohistochemistry reagents were purchased from Biocare Medical (Concord, CA, USA). Immunostainings were performed on tissues from the same mice. Briefly, sections were deparaffinized in Tissue Clear and rehydrated in an ethanol series. Subsequently, slides underwent high-pressure boiling in DIVA buffer (pH 6.0). After blocking with Rodent block (M), sections were incubated with primary antibodies in Da Vinci Green solution for 1 hour at room temperature. Next, a probe-polymer system with alkaline phosphatase was applied, upon which staining was visualized with Warp Red. Slides were counterstained with Hematoxilin QS (Vector Laboratories, Burlingame, CA, USA), dehydrated and mounted in Pertex (Histolab, Gothenburg, Sweden). The following primary antibodies were used: anti-SMA (DAKO, M0851), anti-MYH11 (Abcam, ab53219) and anti-LMOD1 (Sigma, HPA030097).

### 2.8. Statistical Analysis

The UBM and myography data were analyzed by strain in order to investigate the strain-specific flow-mediated remodeling response and presented as mean ± standard error of mean. UBM data was analyzed using 2-way ANOVA with Bonferroni post hoc test for multiple comparisons when comparing differences between groups over time. Comparison of quantifications in electron microscopy and histology, between WT and KO mice, were performed and data presented as mean ± standard deviation. Distribution of data was tested using Sharpio-Wilks normality test. Kruskal-Wallis test with Dunn´s method of comparison and Mann-Whitney U test was used for evaluation of non-paired nonparametric data. Wilcoxon signed rank test was used for analysis of paired nonparametric data over time within a certain group. Correlation coefficient was calculated using Pearson´s test for parametric data and Spearman’s test for nonparametric data. Statistical analysis was performed using two-tails and *p*-value less than 0.05 was considered statistically significant. All statistical analyses were performed using GraphPad Prism 6 (GaphPad Prism Inc., San Diego, CA, USA).

## 3. Results

### 3.1. Reduced Arterial Geometry and Bodyweight in the Pcsk6 KO Mice Under Normal Conditions

Comparison of baseline characteristics between WT and KO mice was performed in order to evaluate the presence of strain-dependent differences, which could potentially interfere with further analysis. A borderline significant decrease in bodyweight of the KO mice with a concomitant significant reduction in diastolic circumference, assessed by ultrasound (Appendix A), was detected. Due to these differences, further analyses in this study were focused on comparing strain-specific controls while between-strain comparisons are given in the Appendix A. 

### 3.2. Increased Outward Remodeling in Pcsk6 KO Mice Exposed to Increased Flow

In order to assess the influence of Pcsk6 on arterial geometry during flow-dependent remodeling, we utilized systematic examinations with ultrasound biomicroscopy. An increase in lumen circumference was detected at 3 and 5 weeks in the KO IF mice compared to strain-specific baseline controls (Table 1, Figure 1A). A similar result could be observed when comparing lumen circumference over time, from intact to 5 weeks after surgery, in the KO mice exposed to increased flow (Figure 1A). In contrast, a non-significant decrease in lumen circumference from 3 to 5 weeks was seen in the WT IF mice (Table 1, Figure 1A). No difference in elasticity in terms of strain could be detected between the IF group and controls (Table 1). However, a flow-mediated increase in strain from intact to 5 weeks after surgery was observed in the WT IF mice (from 18.3 ± 1.0 to 22.5 ± 1.3, *p* < 0.05). Conversion of lumen diameter to circumference did not affect the statistical analysis.

### 3.3. Pcsk6 KO Mice Display a Continuous Increase in Blood Flow in the Contralateral Artery Following Ligation

To investigate the differences in hemodynamics during flow-mediated remodeling in WT and KO mice we analyzed blood flow velocities in combination with arterial geometry. In general, the ligation induced a 45–54% increase in right carotid volume flow rate when compared with strain-specific controls (Table 1). Furthermore, an increase in volume flow rate over time, from 3 to 5 weeks, was seen in the KO IF group (Figure 1B). A flow-mediated increase in EDV at 3 and 5 weeks (*p* < 0.05 and *p* < 0.001) and VTI at 5 weeks (*p* < 0.01) was observed in the WT mice (Table 1). A similar non-significant pattern was seen in KO mice. No statistical difference in mean velocity could be detected.

Comparison of PI revealed a significant decrease 5 weeks after surgery in the Pcsk6 KO mice exposed to increased blood flow. In the WT IF mice, an initial decrease in PI at 3 weeks after surgery was seen. However, due to an increase in PI over time, no statistical difference could be observed at 5 weeks after surgery (Figure 1C). Further analysis revealed a reduction in RI at 3 and 5 weeks after ligation in the WT mice and at 5 weeks in the KO mice (Table 1). Analysis of right carotid WSS could not detect any significant differences between IF or control groups. However, a borderline significant increase in WSS from 3 to 5 weeks (from 104 ± 3.9 to 132 ± 6.2 dynes/cm^2^, *p* = 0.063) was seen in the WT IF mice (Figure 1D).

### 3.4. Presence of PCSK6 Influences Contractility in Flow-Mediated Vascular Remodeling

In order to investigate the influence of Pcsk6 ablation on tensile and contractile characteristics in flow-mediated vascular remodeling (Figure 2A,B) we utilized wire myography. An increased optimal circumference was observed in the KO IF mice (Figure 2A,C). Furthermore, an increased active tension at optimal circumference was detected in the WT IF mice (Figure 2B,D). However, no statistical difference in passive tension at optimal circumference was observed (Table 2). No significant difference in contractile response to phenylephrine in relation to KCl between IF and controls could be observed (data not shown). No difference in length-force curves could be detected between WT and KO mice (data not shown) and no significant effect of IF on thoracic aorta could be identified (Table 2). Of interest, comparison of lumen circumference estimated by myography and UBM revealed a significant correlation and an agreement using the Bland–Altman method of comparison between optimal and diastolic circumference (Figure 3).

### 3.5. Reduced Elastic Laminae Content in Pcsk6 KO Mice

To further investigate the influence of PCSK6 deficiency on arterial morphology upon exposure to increased blood flow we utilized histochemistry and transmission electron microscopy. Weigert-Von Gieson staining showed an overall reduced abundance of elastin in KO mice compared to WT mice. Masson’s trichrome staining revealed that KO mice exposed to IF have increased collagen deposits in the remodeled artery (Figure 4A). Quantification of the number of nuclei in the artery showed that there was no difference in the number of cells per media area (Figure 4B). Transmission electron microscopy revealed a remodeling associated increase in elastic laminae content per media area in the WT mice. In contrast, increased flow did not affect the elastic laminae content in PCSK6 deficient mice, but comparison between WT and KO mice exposed to increased flow revealed a borderline significant trend (Figure 4C,D). No difference in media area could be detected between the groups (Appendix A).

### 3.6. Remodeled Pcsk6 KO Arteries Express Lower Levels of Typical Contractile SMC Markers

In order to understand the cellular and tissue features associated with increased flow-mediated outward remodeling in the Pcsk6 KO mice, immunohistochemistry was performed. Expression of typical contractile SMC markers, such as smooth muscle alpha-actin (SMA), myosin heavy chain 11 (MYH11) and leiomodin-1 (LMOD1) was decreased in KO mice compared to WT mice, upon exposure to IF conditions (Figure 5A,B, Appendix A). Under normal flow conditions, we did not observe major differences in the staining intensity of these markers between KO and WT mice (Figure 5A,B, Appendix A).

## 4. Discussion

In the present study, we reveal that PCSK6 deficiency results in increased outward remodeling and reduced remodeling-associated contractile response in arteries exposed to increased blood flow. Mechanistically, analysis of the arterial wall structure highlighted decreased elastic laminae content and reduced expression of contractile SMC markers in the Pcsk6^−/−^ mice exposed to increased blood flow. These results indicate that PCSK6 is important for modulation of the arterial geometry and SMC contractility in flow-mediated vascular remodeling.

Blood flow measurements verified successful ligation with concomitant increase in volume flow rate in the right CCA, which was in line with previously published studies [32,38]. In the PCSK6 deficient mice we observed a progressive outward remodeling response over time, detected as an increasing arterial circumference in UBM. In comparison, WT mice displayed a non-significant initial luminal expansion at 3 weeks followed by a reduction and normalization at 5 weeks. Previous studies have reported contradicting results concerning the influence of flow-mediated remodeling on arterial geometry in the contralateral artery following complete CCA ligation [30,41,42,43]. In a systematic description of flow-mediated remodeling in the aortic banding model in C57Bl/6J mice, Eberth et al. [44] reported an initial increase followed by an adaptive reduction in arterial geometry over time. Despite the differences in surgical procedure and physiological response, our results reveal a similar temporal pattern with regards to alterations in arterial lumen caliber in the WT mice. Hence, the increased arterial geometry in KO mice suggests that PCSK6 is important for the proper adaptive remodeling response.

Analysis of blood flow velocity, not dependent on circumference or wall thickness, revealed a significant increase in EDV at 3 and 5 weeks in the WT mice subjected to carotid ligation. PI and RI are non-dimensional velocity measurements used to assess organ perfusion in clinical research and to assess vascular impedance and large artery stiffness in experimental cardiovascular research [37,45]. Interestingly, a gradual decrease in PI and RI over time was observed in the Pcsk6^−/−^ mice, which could not be detected in the WTs. The difference in PI and RI was, in part, attributed to an absence of increase in EDV in the KO mice. It should also be taken into account that these findings could be influenced by differences in remodeling response in the distal vasculature or the myocardium, in addition to the specific CCA wall remodeling. Previous studies have sporadically reported on PCSK6-associated cardiac anomalies in humans and a fraction of Pcsk6^−/−^ mice [26,27,28]. However, it has been shown that absence of PCSK6 does not influence the myocardial function under normal conditions and the majority of Pcsk6^-/-^ mice do not carry structural cardiac anomalies [28]. Cardiac anomalies in the KO mice were not particularly addressed in the present study, but it is important to highlight that we did not observe any abnormal blood flow profile or significant differences in bodyweight over time, which suggests absence of major structural anomalies or congestive cardiac failure. In a recent paper it was shown that increased levels of PCSK6 in the post-ischemic myocardium lead to reduced ejection fraction due to increased myocardial fibrosis [46]. However, the full impact of PCSK6 on myocardial remodeling and function remains to be fully elucidated.

The outward remodeling response observed in the PCSK6 deficient mice by ultrasound could also be detected as an increased circumference at optimal stretch in wire myography. Interestingly, a flow-mediated increase in contractility was observed in the WT mice, which could not be detected in the KO mice. Despite the differences in flow-mediated contractility, we could not detect any difference in media thickness in WT compared to the strain specific controls, which is in line with previously published studies [31,38,42]. In the control groups, PCSK6 deficiency did not influence the media thickness or the length-force curves, which indicates that there was no difference in the functional elastic properties of the arterial wall. In a recently published study, we could detect differences in the gene expression of the arterial wall ECM components in Pcsk6^−/−^ mice compared to WT controls [25]. Interestingly, despite this baseline reduction in gene expression which was particularly related to proteoglycans, here we show that the presence of PCSK6 does not seem to influence the functional elastic or contractile properties of the arterial wall under normal physiological conditions, but its role seems to be rather specific for response to alterations in blood flow.

Several histology-based observations validated and extended the results from physiological studies of Pcsk6 KO mice. Transmission electron microscopy revealed a flow-mediated increase in elastic laminae content in the tunica media of WT mice, which could not be detected in KOs. This could be related to a remodeling-associated increase in the collagen-elastin ratio, a common feature in arterial wall stiffening [44]. Therefore, it is possible that the arterial stiffening may have influenced the dilation of the artery upon pressure perfusion, resulting in increased elastic laminae area. Previous studies have suggested an induction of elastogenesis in flow-mediated outward remodeling and vascular diseases [47,48]. Interestingly, the histochemical analysis of the remodeled arteries revealed that there was no difference in the number of SMCs per media area in the Pcsk6 KO mice compared to WT mice. However, a reduced staining for typical markers of contractile SMCs such as SMA, MYH11 and LMOD1 could be observed in the tunica media of KO mice exposed to IF compared to WTs, altogether suggesting the alteration in SMC’s contractile capacity [49]. Hence, the absence of an adaptive response in Pcsk6 KO mice could be related to a reduction in the contractile medial features. Combined with the fact that elastic laminae content was lower in KOs exposed to increased flow, which is shown to be influenced by SMC phenotypic modulation, these findings strongly indicate that PCSK6 is of importance for proper vascular SMC adaptation in flow-mediated vascular remodeling and we speculate that it may also play a role in SMC hypertrophy or contractile vs dedifferentiated cell ratios. PCSK6 has been reported to process and activate growth factors related to vasculogenesis, arterial remodeling and SMC activation, as well as vascular disease, such as TGFB1 and PDGFB [23,24]. TGFB1 stimulates arterial stiffening and has been shown to induce SMC hypertrophy [50,51], while PDGFB signaling has been associated with SMC activation and hypertrophy in vitro [52]. However, the molecular mechanism and role of PCSK6 in growth factor processing and SMC hypertrophy and/or activation in flow-mediated remodeling remains to be elucidated.

Human genetic variants in the PCSK6 locus have been linked with increased blood pressure and aortic dissection [27,53], which are both known to be associated with aortic aneurysm formation [54,55]. The risk for development of aortic dissection and aneurysms is increased in patients with genetic diseases resulting in a defective TGFB-signaling [56], where PCSK6 is one of the key proteases involved in the TGFB1 axis [22,23]. However, whether an altered expression of PCSK6 would be associated with aortic aneurysm disease remains to be further investigated.

### Limitations

In the present study, we did not perform measurements of systemic blood pressure. It has been reported that Pcsk6^−/−^ mice have a 15 mmHg increase in habitual blood pressure compared to WT mice [28]. The influence of PCSK6 deficiency on blood pressure in the carotid ligation model should be further investigated.

The current study is limited in regards to the reduced sample size of each group, which is related to the relatively high embryonic lethality in the Pcsk6^−/−^ mice [22]. In the TEM experiment the control group was limited to two mice per strain. The control animals in this experiment were used to visualize possible major differences in arterial wall anatomy, which could not be detected in either normal or increased flow groups. Due to the consistency of our results throughout the different methodologies, it is unlikely that an increased number of mice per group would alter the results presented in this study.

We could detect PCSK6-dependent differences in morphology, hemodynamics and medial contractility in the model of flow-mediated vascular remodeling. However, the influence of PCSK6 on underlying molecular mechanisms in flow-mediated remodeling remains to be fully characterized. The current study examines the influence of PCSK6 on flow-dependent remodeling in the carotid artery, and whether our findings are applicable to other vascular beds remains to be investigated. In addition, the arterial remodeling process has been shown to be influenced by adventitial remodeling and myofibroblasts, which should also be further elucidated with respect to the role of PCSK6 [37,57].

## 5. Conclusions

Using a combination of physiological and histological methods, we have shown that PCSK6 is of importance for arterial geometry and adaptive contractile response in the model of flow-mediated vascular remodeling, mechanistically via modulation of SMC contractile markers. These findings deepen our understanding of pathological remodeling mechanisms in vascular disease and after vascular interventions.

## Figures and Tables

**Figure 1 cells-09-01009-f001:**
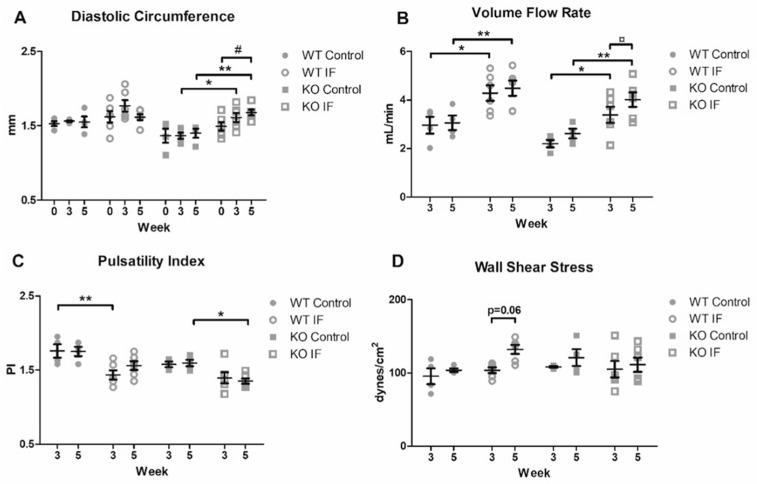
Morphology and hemodynamics of arteries exposed to increased flow measured by ultrasound biomicroscopy. Analysis of the right CCA revealed an increased flow-mediated (**A**) outward remodeling and (**B**) volume flow rate over time in the KO mice. A different pattern in (**C**) PI was seen between WT and KO mice. No significant difference in (**D**) wall shear stress could be detected. 2-way ANOVA with Bonferroni Multiple Comparison test was used for comparing differences between strain-specific mice under normal (WT *n* = 4, KO *n* = 4) and IF (WT *n* = 6, KO *n* = 6) conditions at the same time point, * *p* < 0.05 and ** *p* < 0.01. Wilcoxon signed rank test was used for comparing data within the same group from 3 to 5 weeks, ¤ *p* < 0.05, and 0 to 5 weeks, # *p* < 0.05. Data are expressed as mean ± standard error of mean. CCA = common carotid artery. IF = increased blood flow. KO = Pcsk6^−/−^ mice. PI = Pulsatility index. WT = C57Bl/6J mice.

**Figure 2 cells-09-01009-f002:**
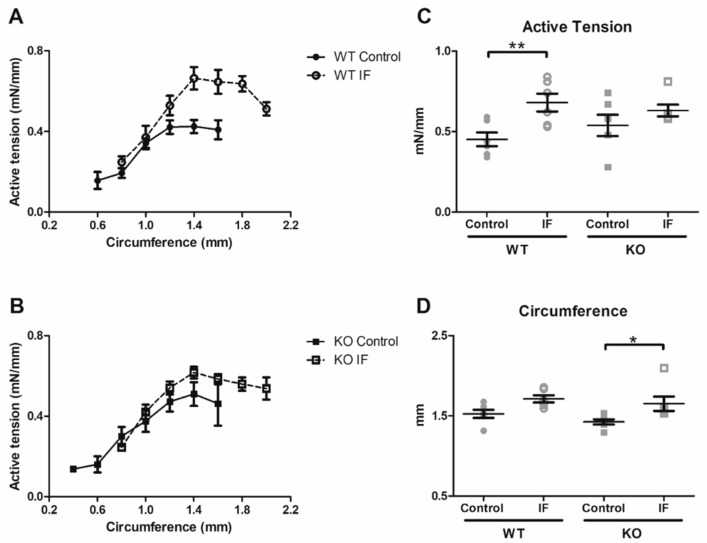
Impact of PCSK6 on vascular physiology measured by wire myography. Measurements of active tension at increasing circumferences categorized in intervals of 0.2 mm in (**A**) WT (control *n* = 4, IF *n* = 6) and (**B**) KO (control *n* = 4, IF *n* = 6) mice. Comparison to controls revealed an increase in (**C**) active tension at (**D**) optimal circumference in the WT mice exposed to increased blood flow, which was not observed in the KO mice. * *p* < 0.05, ** *p* < 0.01 calculated using Kruskal–Wallis test with Dunn´s test for multiple comparison. Data expressed as mean ± standard error of mean. CCA = common carotid artery. IF = increased blood flow. KO = Pcsk6^−/−^ mice. WT = C57Bl/6J mice.

**Figure 3 cells-09-01009-f003:**
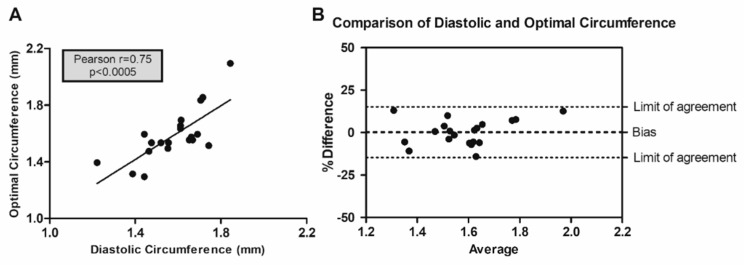
Comparison of the right common carotid artery circumference measured by wire myography and ultrasound biomicroscopy. A significant correlation was detected when comparing circumferences (**A**), measured in diastolic phase by ultrasound biomicroscopy and at optimal stretch in wire myography. (**B**) Bland-Altman method of comparison, visualized as percent difference, revealed an agreement between the methods (bias = 0.27%, standard deviation = 7.63%). The analysis includes data from Pcsk6 KO and WT mice under normal (WT *n* = 4, KO *n* = 4) and increased flow (WT *n* = 6, KO *n* = 6) conditions. KO = Pcsk6^−/−^ mice. WT = C57Bl/6J mice.

**Figure 4 cells-09-01009-f004:**
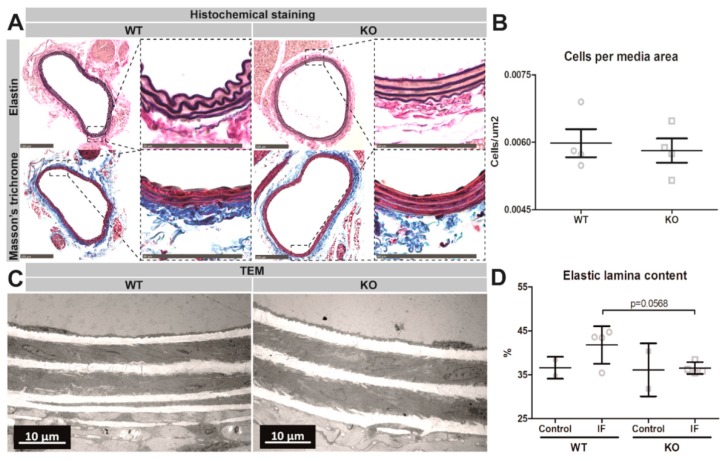
The influence of PCSK6 on cell density and elastic laminae content in the right common carotid artery. Representative images from WT and KO carotid arteries exposed to increased flow using histochemical staining with (**A**) Weigert’s elastin and Masson’s trichrome (WT *n* = 4, KO *n* = 4). Images from Masson’s trichrome staining were used to (**B**) quantify the number of nuclei per media area in the outward remodeled vessels. Further evaluation was performed using transmission electron microscopy on pressure-fixed right CCAs 6 weeks after exposure to increased blood flow in (**C**) WT and KO mice. Quantification was performed by measurements of (**D**) elastic laminae content per media area during normal (WT *n* = 2, KO *n* = 2) and increased flow (WT *n* = 4, KO *n* = 4) conditions. Data expressed as mean ± standard deviation. * *p* < 0.05 calculated with Mann-Whitney test. CCA = common carotid artery. IF = increased blood flow. KO = Pcsk6^−/−^ mice. TEM = transmission electron microscopy, WT = C57Bl/6J mice.

**Figure 5 cells-09-01009-f005:**
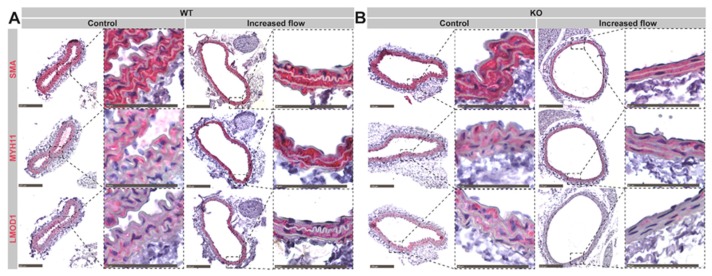
The influence of PCSK6 deficiency on the expression of typical smooth muscle cell markers upon increased flow in the right common carotid artery. Representative images of the carotid arteries stained for smooth muscle alpha-actin (SMA), myosin heavy chain 11 (MYH11) and leiomodin-1 (LMOD1) from (**A**) WT and (**B**) KO mice under normal (WT *n* = 2, KO *n* = 4) and increased flow conditions (WT *n* = 4, KO *n* = 4). KO = Pcsk6^−/−^ mice. WT = C57Bl/6J mice.

**Table 1 cells-09-01009-t001:** Ultrasound biomicroscopy assessment of vascular remodeling in the right CCA following left carotid ligation.

	WT				KO			
	Control		Increased flow		Control		Increased flow	
	3 weeks	5 weeks	3 weeks	5 weeks	3 weeks	5 weeks	3 weeks	5 weeks
Systolic circumference (mm)	1.94 ± 0.03	1.89 ± 0.06	2.09 ± 0.06	1.99 ± 0.05	1.67 ± 0.04	1.72 ± 0.07	1.95 ± 0.06 *	2.02 ± 0.04 **
Diastolic circumference (mm)	1.56 ± 0.01	1.55 ± 0.07	1.77 ± 0.08	1.62 ± 0.04	1.37 ± 0.04	1.40 ± 0.06	1.61 ± 0.06 *	1.68 ± 0.04 **
Elasticity in terms of strain (%)	23.6 ± 1.1	22.1 ± 1.3	18.4 ± 2.4	22.5 ± 1.3	22.4 ± 2.6	22,2 ± 1.0	21.1 ± 2.4	19.7 ± 0.9
Peak systolic velocity (mm/s)	692 ± 96.6	807 ± 70.0	630 ± 45.7	828 ± 47.8	564 ± 31.2	687 ± 64.2	553 ± 54.2	603 ± 59.1
End diastolic velocity (mm/s)	105 ± 12.6	108 ± 4.7	143 ± 6.6 *	169 ± 9.8 ***	115 ± 4.9	126 ± 4.3	139 ± 14.2	157 ± 12.1
Mean velocity (mm/s) (mm/s)	332 ± 46.3	397 ± 30.7	336 ± 19.4	422 ± 20.0	284 ± 13.9	349 ± 31.6	296 ± 29.6	329 ± 32.2
Velocity-time integral (mm)	27.2 ± 1.9	29.5 ± 1.3	33.4 ± 0.9	38.4 ± 1.6 **^,¤^	29.2 ± 0.8	32.3 ± 2.4	35.2 ± 2.2	37.7 ± 2.4
Volume flow rate (mL/min)	2.96 ± 0.35	3.01 ± 0.30	4.28 ± 0.32 *	4.49 ± 0.31 **	2.20 ± 0.15	2.62 ± 0.20	3.39 ± 0.33 *	4.02 ± 0.30 **^,¤^
Resistive Index (RI)	0.84 ± 0.02	0.86 ± 0.02	0.77 ± 0.02 *	0.79 ± 0.01 *	0.80 ± 0.02	0.81 ± 0.02	0.75 ± 0.02	0.74 ± 0.01 *

Morphological and hemodynamic measurement by ultrasound biomicroscopy at different time points after carotid ligation. Data are presented as mean ± standard error of mean and includes data from normal (WT *n* = 4, KO *n* = 4) and increased flow (WT *n* = 6, KO *n* = 6) conditions. * *p* < 0.05, ** *p* < 0.01 and *** *p* < 0.001 compared to strain-specific control at the same time-point using 2-way ANOVA with Bonferroni multiple comparison test. ^¤¤^
*p* < 0.05 when comparing a parameter within the same group from 3 to 5 weeks using Wilcoxon signed rank test. CCA = common carotid artery. KO = Pcsk6^−/−^ mice. WT = C57Bl/6J mice.

**Table 2 cells-09-01009-t002:** Wire myography parameters.

	WT		KO	
	Control	Increased flow	Control	Increased flow
Right CCA				
Active tension (mN/mm)	0.45 ± 0.04	0.68 ± 0.06 *	0.54 ± 0.07	0.63 ± 0.04
Passive tension (mN/mm)	2.81 ± 0.15	3.36 ± 0.21	2.82 ± 0.21	3.38 ± 0.28
Circumference (mm)	1.52 ± 0.05	1.71 ± 0.04	1.42 ± 0.03	1.65 ± 0.09 *
Thoracic aorta				
Active tension (mN/mm)	1.36 ± 0.21	1.59 ± 0.10	1.53 ± 0.09	1.73 ± 0.12
Passive tension (mN/mm)	6.90 ± 0.83	7.63 ± 0.42	6.04 ± 0.77	7.68 ± 0.39
Circumference (mm)	3.64 ± 0.25	3.71 ± 0.07	3.51 ± 0.18	3.53 ± 0.09

Comparison between ligated and control in WT (control *n* = 4, IF *n* = 6) and KO (control *n* = 4, IF *n* = 6) mice using Kruskal-Wallis test with Dunn’s test for multiple comparison, *****
*p* < 0.05. Data expressed as mean ± standard error of mean. CCA = common carotid artery. KO = Pcsk6^−/−^ mice. WT = C57Bl/6J mice.

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
