# Peer review of "Lack of PCSK6 Increases Flow-Mediated Outward Arterial Remodeling in Mice"

_cells, 2020, doi:10.3390/cells9041009_

Round 1
Reviewer 1 Report
In “Lack of PCSK6 increases flow-mediated outward arterial remodeling in mice” the authors showed that the proprotein convertase PCSK6, or PACE4, plays a role in arterial remodeling by combining KO of the PCSK6 gene and an animal model of flow-dependent vascular remodeling that utilizes carotid artery ligation. Their physiological and histological data clearly showed that the absence of PCSK6 influences how the vasculature adapts to increased blood flow. The experimental approach and the quality of the data seem appropriate. This reviewer feels that still, the presentation needs significant improvements especially in two aspects:
1) There seems to be a gap in the logic progression as the effect of the PCSK6 -/- KO in the absence of increased blood flow was not formally included. This issue needs to be addressed first before moving into faster blood flow conditions, probably with an opening section dedicated to compare the WT and KO controls. In figure 1, panels A and B show differences between WT and KO controls that seem statistically significant. However, only the differences of IF between the WT and KO group was analyzed. In panels B, C, and D, the data at time 0 was not included. Even if there were no differences, data at time 0 should be included to support the strength of the experimental approach. The same is true to panels B and D in figure 2, the controls should be compared. Panels A and C probably could be combined as the most relevant result is the difference between WT IF and KO IF. Data in tables 1 and 2 probably should be revised to highlight any significant statistical differences between the two controls.
Any information in the literature and from this study about the effects of PCSK6 -/- KO should be combined and discussed in order to highlight the reason why applying a flow model is justified.
2) The writing does not stand alone, especially in the Results section, it is hard to follow. I had to analyze the data in the figures and tables in order to understand what was in the writing. Many paragraphs and sentences start with “using the technique A we collected data that means that B affects C”. In order to keep with the logic, it is better to write sentences that start with “in order to answer question A, we tested the effect of B by measuring C utilizing the method D”. Presenting the results as a series of questions helps the reader with following the logic.
Another problem I encountered was the substitution of terms that denote experimental variables with other words. For example, “increased blood flow” is the experimental variable, and it is frequently substituted with “carotid ligation” or “surgery”. Although the last two expressions describe the procedures required to obtain the experimental variable, their use in the writing is confusing and does not help with the following of the logic.
Author Response
Reviewer #1: Comments and Suggestions for Authors
“In “Lack of PCSK6 increases flow-mediated outward arterial remodeling in mice” the authors showed that the proprotein convertase PCSK6, or PACE4, plays a role in arterial remodeling by combining KO of the PCSK6 gene and an animal model of flow-dependent vascular remodeling that utilizes carotid artery ligation. Their physiological and histological data clearly showed that the absence of PCSK6 influences how the vasculature adapts to increased blood flow. The experimental approach and the quality of the data seem appropriate. This reviewer feels that still, the presentation needs significant improvements especially in two aspects:”
Response
The authors thank the Reviewer for acknowledging our research and for the insightful comments, which have been addressed as requested and have indeed improved our manuscript.
“1) There seems to be a gap in the logic progression as the effect of the PCSK6 -/- KO in the absence of increased blood flow was not formally included. This issue needs to be addressed first before moving into faster blood flow conditions, probably with an opening section dedicated to compare the WT and KO controls. In figure 1, panels A and B show differences between WT and KO controls that seem statistically significant. However, only the differences of IF between the WT and KO group was analyzed. In panels B, C, and D, the data at time 0 was not included. Even if there were no differences, data at time 0 should be included to support the strength of the experimental approach. The same is true to panels B and D in figure 2, the controls should be compared. Panels A and C probably could be combined as the most relevant result is the difference between WT IF and KO IF. Data in tables 1 and 2 probably should be revised to highlight any significant statistical differences between the two controls. Any information in the literature and from this study about the effects of PCSK6 -/- KO should be combined and discussed in order to highlight the reason why applying a flow model is justified.”
Response 1)
The Reviewer has raised a relevant point in regards to the differences in baseline characteristics between PCSK6 KO and WT mice. At baseline there was a borderline significant difference in bodyweight (p=0.06) between the strains and also a significant difference in diastolic lumen circumference (p=0.009). Due to the difference in arterial geometry, which is also included in calculations for physiological parameters such as volume flow rate, we chose to restrict the analysis to the influence of increased blood flow in a strain-specific manner. In order to clarify on this issue, we have now added a section to the results (p. 5-6, lines 240-246) and also added inter-strain comparisons in Supplementary tables 1 and 2 (page 19). Unfortunately, we did not collect data regarding blood flow velocity at time 0 (Fig 1B-D). The authors agree that this data would have strengthened our manuscript.
“3.1. Reduced arterial geometry and bodyweight in the Pcsk6 KO mice under normal conditions
Comparison of baseline characteristics between WT and KO mice was performed in order to evaluate presence of strain-dependent differences, which could potentially interfere with further analysis. A borderline significant decrease in bodyweight of the KO mice with a concomitant significant reduction in diastolic circumference, assessed by ultrasound (Supp. Table 1), was detected. Due to these differences, further analyses in this study were focused on comparing strain-specific controls while between-strain comparisons are given in the Supplementary. “
The Reviewer has insightfully brought to our attention that the effects of PCSK6-/- KO and usage of the increased blood flow model in this study should be elaborated in more detail. As we have mentioned in the Introduction “The influence of PCSK6 on cardiovascular system and disease development still remains largely unknown, although variants in the PCSK6 genomic locus have been associated with congenital heart disease and aortic dissection [2,3] and PCSK6 has been implicated as a regulator of blood pressure in mice subjected to sodium-chloride enriched diet [4]. “ Although there is 25% embryonic lethality in this strain due to improper TGFb1 and PDGFB activation/processing, adult PCSK6 mice appear physiologically normal and without overt cardiovascular phenotypes [1]. We have recently shown that PCSK6 levels in the normal vasculature under baseline conditions are relatively low in both human and murine tissues, and become strongly upregulated in pathological conditions or upon vascular challenge [5]. Despite the relatively low expression levels, in that study we have performed microarray profiling to identify any global gene expression perturbations in the Pcsk6-/- vs. WT carotid arteries. Overall, we detected substantial gene expression differences (about 2000 dysregulated genes) and interestingly, we found that there was a trend towards downregulation in the KO arteries of many Pdgf and Tgf cytokines including their receptors. There was also a downregulation of SMC contractile markers and many ECM molecules (especially those related to proteoglycans), and regulation of cell death/survival pathways was perturbed. Thus, we concluded that even at baseline KO arteries have lower capacity for expression, as well as reaction to, cytokines and mitogens needed for a complete SMC response to vascular injury. Medial SMCs in the KOs have less contractile and ECM-secretory capacity at baseline and are prone to senescence, indicating that the media likely has reduced functionality. Histologically, in that study we have not observed any major structural differences in arteries from Pcsk6-/- mice compared to controls at baseline. The lack of an obvious vascular phenotype in adult Pcsk6-/- mice indicates the existence of complex compensatory mechanisms, although these are clearly not enough to fully restore the vascular function upon exposure to challenge.
This explains why we have not conducted a more detailed baseline cardiovascular and blood pressure phenotyping of the mice in this study, as we have clearly stated in the Discussion “However, it has been shown that absence of PCSK6 does not influence the myocardial function under normal conditions and the majority of Pcsk6-/- mice do not carry structural cardiac anomalies [4]. Cardiac anomalies in the KO mice were not particularly addressed in the present study, but it is important to highlight that we did not observe any abnormal blood flow profile or significant differences in bodyweight over time, which suggests absence of major structural anomalies or congestive cardiac failure.” Nevertheless, the PCSK6 gene variants were previously linked to hypertension in humans [6] and recent publications have already shown that Pcsk6 KO mice were hypertensive on a normal-salt (0.3% NaCl) diet [4]. The hypertensive phenotype was exacerbated when the mice were placed on high-salt (4% and 8% NaCl) diets, indicating that PCSK6-mediated Corin activation in the heart is essential for sodium homeostasis and normal blood pressure in mice and that PCSK6 deficiency may cause salt-sensitive hypertension in humans and even implication in heart failure [4]. Consistent with these findings, another most recent study showed an increase of PCSK6 levels in serum from patients with acute myocardial infarction, with a peak on day 3 postinfarction. Mechanistically, PCSK6 levels were elevated in hypoxic cardiomyocytes and in hearts of mice following ligation of the left anterior descending artery in the same study [7]. The Authors concluded that enrichment of PCSK6, similarly as shown in our recent study with respect to atherosclerotic or injured vasculature [5], could be crucially involved in cardiac remodeling after acute myocardial infarction. We thank the Reviewer for instigating us to expand on this background, which has now been added to the revised manuscript Introduction (page 2, line 77-80).
“Interestingly, deletion of Pcsk6 does not cause any obvious vascular phenotype detectable in adult Pcsk6-/- mice despite some alterations in the overall gene expression, which indicates that compensatory mechanisms are present. However, we have previously shown that these compensatory mechanisms are insufficient to uphold the normal function of medial SMCs in response to alterations in the physiological milieu [5].”
“2) The writing does not stand alone, especially in the Results section, it is hard to follow. I had to analyze the data in the figures and tables in order to understand what was in the writing. Many paragraphs and sentences start with “using the technique A we collected data that means that B affects C”. In order to keep with the logic, it is better to write sentences that start with “in order to answer question A, we tested the effect of B by measuring C utilizing the method D”. Presenting the results as a series of questions helps the reader with following the logic.
Another problem I encountered was the substitution of terms that denote experimental variables with other words. For example, “increased blood flow” is the experimental variable, and it is frequently substituted with “carotid ligation” or “surgery”. Although the last two expressions describe the procedures required to obtain the experimental variable, their use in the writing is confusing and does not help with the following of the logic.”
Response 2)
We thank the Reviewer for these comments and we have now adjusted the manuscript according to the Reviewers suggestion.
Reviewer 2 Report
In this manuscript, the authors demonstrate flow-mediated remodeling in Pcsk6 knockout mice. This is a logical follow-up of their previous work and provides a comprehensive overview of the effects of a loss of Pcsk6. Overall the work is interesting and mechanistic data is robust.
Specific comments
1. Are hearts from PCSK6 knockout mice already impaired? Previous work has shown an absence of structural abnormalities, however, figure 1 and table 1 suggest a reduction in Diastolic circumference, volume flow rate, etc. are these significant between WT control and KO control?
2. The morphology of the aorta appears altered in the IF KO model a) would this affect any of the analysis ie. Lumen diameter. b) are there any possible implications of the smooth aortic tone?
3. It would be interesting to determine when the proliferation of cells (Figure 4, B) occurs. Does this precede the changes seen in the heart parameters as expected or is this a consequence of the changes in other physiological parameters?
4. Figure 4, B states that 2 WT mice were used to quantify the number of nuclei per media, however, 4 points can be seen on the graph. Further clarification of this is necessary. Are two points taken from the same WT animal?
5. In the methods, it should state what statistical analysis software was used.
6. For the figures legends, please state the number of mice used.
7. In discussion ‘Also, reduced staining for typical markers of contractile SMCs such as SMA, MYH11, and LMOD1 could be observed in the KO mice media compared to WTs, altogether suggesting the presence of dedifferentiated SMCs [48]’ It should be made clear that this is referring to mice with IF. The definition of AVF is missing.
Author Response
Reviewer #2: Comments and Suggestions for Authors
“In this manuscript, the authors demonstrate flow-mediated remodeling in Pcsk6 knockout mice. This is a logical follow-up of their previous work and provides a comprehensive overview of the effects of a loss of Pcsk6. Overall the work is interesting and mechanistic data is robust.”
On behalf of all Authors, we wish to thank the Reviewer for acknowledging our study and for the intricate comments which have indeed helped us to improve our manuscript.
Specific comments
“1. Are hearts from PCSK6 knockout mice already impaired? Previous work has shown an absence of structural abnormalities, however, figure 1 and table 1 suggest a reduction in Diastolic circumference, volume flow rate, etc. are these significant between WT control and KO control?”
Response 1)
We thank the Reviewer for bringing to our attention that further clarification on this subject is needed. In a previous study, it was shown that the Pcsk6 KO mice without major cardiac malformations do not have an impaired cardiac function [4]. However, we could detect a borderline significant difference in baseline bodyweight with concomitant significant differences in diastolic circumference. Due to these differences we chose to analyze the data using strain-specific controls rather than performing an inter-strain analysis. In order to respond to the Reviewer’s questions, the inter-strain baseline data has been added to the revised manuscript as Supplementary Tables 1 and 2. Also, we have added a more detailed description of the cardiac function to the Introduction (page 2, line 75-77). Please also see our Response 1) to Reviewer #1.
“In comparison to wild-type controls, Pcsk6-/- mice have been shown to have a slight increase in habitual systolic blood pressure without alterations in myocardial thickness or function [4].”
“2. The morphology of the aorta appears altered in the IF KO model a) would this affect any of the analysis ie. Lumen diameter. b) are there any possible implications of the smooth aortic tone?”
Response 2)
We thank the Reviewer for bringing this to our attention. In the current manuscript, the histological staining was performed on carotid arteries without pressure fixation. Therefore, any conclusions regarding morphological differences should be made with caution. However, we did perform transmission electron microscopy on pressure fixed carotid arteries, which did not reveal any major structural difference in the arterial wall in either normal or increased flow conditions. These results are in line with our previously published study [5]. For clarification, we have now specified vessel names in the caption of figures containing histological sections.
In this study arterial function between the WT and KO mice was investigated using wire myography. We did not detect any significant difference in active tension in the arterial wall of KO mice exposed to IF in neither aorta nor carotid artery. Interestingly, the circumference at optimal stretch in wire myography corresponded well with the diastolic circumference measured in ultrasound (Fig 3A-B), indicating that the wire myography analysis can be regarded as a valid ex vivo representation of the physiological milieu in vivo.
However, we have in our previous study [5] observed some baseline differences on the global gene expression level between carotid arteries from KO vs WT mice. Thus, it may be that, although the vessel functionality and gross morphology is not impaired at baseline, KO arteries have a lower capacity for expression, as well as reaction to, cytokines and mitogens needed for a complete SMC response to vascular challenge, and that those pathways may have implications for the effects seen under the IF model here.
“3. It would be interesting to determine when the proliferation of cells (Figure 4, B) occurs. Does this precede the changes seen in the heart parameters as expected or is this a consequence of the changes in other physiological parameters?”
Response 3)
The Reviewer highlights an interesting and valid question. The influence of increased blood flow on arterial wall cell density has been systematically investigated in the aortic banding model, in which a constrictive banding is performed on the aortic arch distal to the innominate artery, also known as the brachiocephalic trunk [8]. Despite the differences in methodology, with aortic banding causing greater increase in flow through the right common carotid artery, Eberth et al showed that arterial wall cell density increases as early as 7 days after flow increase [8]. Using a modified carotid ligation model, in which the left common carotid artery flow is diverted to the left superior thyroid artery, Sullivan and Hoying showed that arterial wall cell proliferation and apoptosis indices were both increased already at 4 days after surgery [9]. The exact time point for induction of cell proliferation and its relation to physiological parameters remains to be fully elucidated.
The current study was designed primarily to investigate if deletion of Pcsk6 would influence arterial physiology and morphology upon increased blood flow, rather than describing the temporal differences throughout the remodeling process. Therefore, our study design did not include groups for euthanization and tissue harvest at different time points. The Authors agree with the Reviewer that it would be of interest to fully describe the temporal effects of flow-induced arterial remodeling on cell proliferation and its relation to arterial and cardiac physiology. However, such an approach is out of scope for the current study.
“4. Figure 4, B states that 2 WT mice were used to quantify the number of nuclei per media, however, 4 points can be seen on the graph. Further clarification of this is necessary. Are two points taken from the same WT animal?”
Response 4)
We thank the Reviewer for bringing this inconsistency to our attention. Upon addressing this issue, a misclassification and an error with scale-setting was discovered, which led us to repeat the image quantifications from this experiment. Our new results reveal that there are no significant differences in cells per media area between the different strains. We apologize for this mistake which we have corrected now in Figure 4B and appropriate changes have been made throughout the manuscript (page 3, line 98-99, 102,118; page 5, line 205; page 10, line 331; page 13, line 424-434).
“Quantification of the number of nuclei in the artery showed that there was no difference in number of cells per media area (Figure 4B).”
“Interestingly, the histochemical analysis of the remodeled arteries revealed an that there was no difference in number of SMCs per media area in the Pcsk6 KO mice compared to WT mice. However, a reduced staining for typical markers of contractile SMCs such as SMA, MYH11 and LMOD1 could be observed in the tunica media of KO mice exposed to IF compared to WTs, altogether suggesting the alteration in SMCs contractile capacity [10]. Hence, the absence of adaptive response in Pcsk6 KO mice could be related to a reduction in the contractile medial features. Combined with the fact that elastic laminae content was lower in KOs exposed to increased flow, which is shown to be influenced by SMC phenotypic modulation, these findings strongly indicate that PCSK6 is of importance for proper vascular SMC adaptation in flow-mediated vascular remodeling and we speculate that it may also play a role in SMC hypertrophy or contractile vs dedifferentiated cell ratios.”
“5. In the methods, it should state what statistical analysis software was used.”
Response 5)
The statistical software with reference has now been added (page 5, line 237-238).
“All statistical analyses were performed using GraphPad Prism 6 (GaphPad Prism Inc., San Diego, CA, USA).”
“6. For the figures legends, please state the number of mice used.”
Response 6)
The number of mice in each figure legend has been added.
“7. In discussion ‘Also, reduced staining for typical markers of contractile SMCs such as SMA, MYH11, and LMOD1 could be observed in the KO mice media compared to WTs, altogether suggesting the presence of dedifferentiated SMCs [48]’ It should be made clear that this is referring to mice with IF. The definition of AVF is missing.”
Response 7)
The authors thank the Reviewer for addressing these issues. We have now corrected the manuscript with inclusion of AVF definition (page 1, line 41) and clarification of which group of mice is referred to (page 13, line 427).
“..with arteriovenous dialysis fistulas, due to an excessive shunting..”
“Also, a reduced staining for typical markers of contractile SMCs such as SMA, MYH11 and LMOD1 could be observed in the tunica media of KO mice exposed to IF compared to WTs, altogether suggesting the presence of dedifferentiated SMCs [10].”
Reviewer 3 Report
The authors examine the role of Pcsk6 during arterial wall remodeling in response to increased blood flow and use for this purpose an established model of left carotid artery ligation and examine the contralateral site and Pcsk6 deficient mice. They find that systemic absence of Pcsk6 is associated with SMC dedifferentation and loss of elastic fibers and results in increase arterial vessel diameter and dilatation as well as increase in flow.
The data are interesting, the methodology state-of-the art. The paper is well-written, although some parts of the Introduction could be shortened. I also appreciate that the methodology was well described.
In addition, I have the following specific comments:
- Differences in vessel diameter, flow rate, pulsatility index and shear rate between WT and Pcsk6 KO mice were also observed at baseline (Figure 1 and 2A). Were these differences between strains significant? Non-significant differences are not indicated in the graphs, therefore it is not clear whether this comparison was performed or not. In this regard, some columns without indication of significant differences look similar to those which significantly differed. Please clarify.
- Were differences in lumen diameter and flow at baseline also present in other vascular beds (e.g. aorta) or was this not examined. Please mention in the text.
- Also regarding the baseline differences and the fact that C57Bl6J control mice were not littermates. Please clearly indicate the genetic background of the PSCK6-/- mice and if and for how many generations they were backcrossed. Please also indicate the sex and age of mice, as both are relevant for the vascular remodeling response and may affect Pcsk6 expression.
- Which cells in the arterial wall primarily express PCSK6, before and after injury? The authors focus on SMCs, but what about endothelial cells, an important cell type in the control of vascular tone. Did the authors observe signs of endothelial activation?
- Figure 4: the number of cells is expressed per media area. Was the media area itself similar or different between strains?
- Figure 5: Although the images are quite convincing, a quantitative analysis of the representative images should be provided.
- Mechanisms underlying the observations not clear: how does absence of Pcsk6 alter SMC phenotype and cell number? The authors discuss increased proliferation, but decreased apoptosis or better survival should also be mentioned.
- Lastly, can the authors speculate about clinical situations in which reduced Pcsk6 expression may play a role. For example, age or aneurym formation?
Author Response
Reviewer #3: Comments and Suggestions for Authors
“The authors examine the role of Pcsk6 during arterial wall remodeling in response to increased blood flow and use for this purpose an established model of left carotid artery ligation and examine the contralateral site and Pcsk6 deficient mice. They find that systemic absence of Pcsk6 is associated with SMC dedifferentation and loss of elastic fibers and results in increase arterial vessel diameter and dilatation as well as increase in flow.
The data are interesting, the methodology state-of-the art. The paper is well-written, although some parts of the Introduction could be shortened. I also appreciate that the methodology was well described.”
On behalf of all Authors, we thank the Reviewer for the positive comments about our study and for giving us the opportunity to further improve the manuscript.
“In addition, I have the following specific comments:
- Differences in vessel diameter, flow rate, pulsatility index and shear rate between WT and Pcsk6 KO mice were also observed at baseline (Figure 1 and 2A). Were these differences between strains significant? Non-significant differences are not indicated in the graphs, therefore it is not clear whether this comparison was performed or not. In this regard, some columns without indication of significant differences look similar to those which significantly differed. Please clarify.”
Response 1)
We thank the Reviewer to addressing this issue. The similar question has been raised by Reviewer #1 (Response 1) and by Reviewer #2 (Response 1). The above mentioned issues have been addressed, we now added a section to the results (p. 5-6, lines 240-246) and also added inter-strain comparisons in Supplementary tables 1 and 2 (page 19). Unfortunately, we did not collect data regarding blood flow velocity at time 0 (Fig 1B-D).
- “Were differences in lumen diameter and flow at baseline also present in other vascular beds (e.g. aorta) or was this not examined. Please mention in the text.”
Response 2)
Ultrasound examinations were only performed on right and left common carotid arteries. In order to clarify on this subject, we have added the following sentence to the “Limitations” section of our discussion (page 14, line 462-464).
“The current study examines the influence of Pcsk6 on flow-dependent remodeling in the carotid artery, whether our findings are applicable to other vascular beds remains to be investigated.”
- “Also regarding the baseline differences and the fact that C57Bl6J control mice were not littermates. Please clearly indicate the genetic background of the PSCK6-/- mice and if and for how many generations they were backcrossed. Please also indicate the sex and age of mice, as both are relevant for the vascular remodeling response and may affect Pcsk6 expression.”
Response 3)
We apologize for not stating the age and sex of the animals. Age-matched male wild-type (WT) C57BL/6J control mice and Pcsk6-/- mice (5-6 months old) were used for all experiments and this information has been added to the manuscript (page 3, line 98-99). The information regarding the genetic background is already mentioned in the Method part of the manuscript (page 3, lines 97-98).
“A total of 46 male animals with an age of 5-6 months were used in this study.”
- “Which cells in the arterial wall primarily express PCSK6, before and after injury? The authors focus on SMCs, but what about endothelial cells, an important cell type in the control of vascular tone. Did the authors observe signs of endothelial activation?”
Response 4)
We thank the Reviewer for this relevant question. In a previous study we demonstrated that PCSK6 is strongly upregulated in human atherosclerotic carotid plaques and is primarily expressed by smooth muscle cells (SMC), inflammation, extracellular matrix remodeling and mitogens [11]. More recently, we utilized a systems biology approach to further understand the role of PCSK6 in normal and pathological conditions in the vasculature. We revealed that PCSK6 is a novel protease which induces SMC migration in response to PDGFB stimulation, via modulation of contractile markers and MMP14 activation [5]. We have also noted that the PCSK6 expression pattern in plaques is localized to neovessels, but it was not prominent in luminal or neovessel endothelial cells. Also, the expression of PCSK6 in both human and murine vasculature is at baseline relatively low and mainly restricted to the medial layer. Together, our previous studies have established PCSK6 as a key regulator of SMC function in vascular remodeling [5,11]. Although we do not claim that PCSK6 expression is restricted only to vascular SMCs, we have not yet thoroughly investigated the other possible cellular sources of PCSK6 and we so far have no evidence of its’ expression in the endothelium.
- “Figure 4: the number of cells is expressed per media area. Was the media area itself similar or different between strains?”
Response 5)
The data in Figure 4 has been revised upon the comment from Reviewer #2. In direct response to this question, we did not detect any difference in media area between the WT and KO mice in either histomorphometry or transmission electron microscopy. The results from this analysis is now included as Supplementary Figure 1 (page 19).
- “Figure 5: Although the images are quite convincing, a quantitative analysis of the representative images should be provided.”
Response 6)
Unfortunately, the experiment shown in Figure 5 was not performed as a quantitative but rather as a qualitative experiment, in order to examine the differences in expression of contractile smooth muscle cell markers between the genotypes under increased flow conditions. In order to accommodate for the Reviewer’s request, we have chosen to show the images of several animals under increased flow conditions in Supplementary figure 2 (page 20). We would also like to refer to our previous publication where we show in Supplementary Figure V that at baseline level there is already a decreased transcriptomic expression of some typical smooth muscle cell markers between KO and WT animals, such as Myh11 and Lmod1 [5]. However, on protein level assessed by immunohistochemistry stainings, at baseline these differences were not obvious and they could be observed only under increased flow conditions (Figure 5).
- “Mechanisms underlying the observations not clear: how does absence of Pcsk6 alter SMC phenotype and cell number? The authors discuss increased proliferation, but decreased apoptosis or better survival should also be mentioned.”
Response 7)
The Reviewer is correct in raising this issue. During the revision process the analysis with quantification of cells per media area has been repeated due to errors in the scale-setting and a misclassification within our dataset. The new data reveal that there are no significant differences in cells per media area between KO and WT mice. We apologize for having to revise this data and in order to accommodate for these new results we have made changes in the manuscript, please see Response 4) Reviewer #2.
It is nevertheless relevant to speculate on the mechanisms underlying the SMC observations under IF, as the Reviewer suggests. With respect to the new data, we have modified our Discussion (page 13, line 424-440) to indicate a role for PCSK6 in SMC hypertrophy and/or contractile vs dedifferentiated cell ratios, which seem as the most plausible speculation in combination with the effects that we have described when it comes to the reduction of SMC contractility markers, lower Elastin content and an increase in collagen content in Pcsk6 KO arteries upon increased flow, yet no difference in medial cell numbers between KOs and WTs. This is also an interesting possibility since PCSK6 has been shown to process and activate growth factors TGFB1 and PDGFB, which both have been shown to stimulate arterial stiffening, SMC hypertrophy and activation. We have also already referred in the Discussion (page 13, lines 411-416) to our recent publication where we have studied these mechanisms in detail coupled to the models of rat intimal hyperplasia formation after balloon injury, mouse carotid ligation and in human atherosclerosis. In that study we have observed some differences on the gene expression level, as medial SMCs in the KOs have less contractile and ECM-secretory capacity [5]. These differences are not sufficient to be observable on the protein level by immunohistochemistry or even functionally by myography examination performed in this study, but they indicate that KO arteries may have an impaired capacity for a complete medial response upon vascular challenge. Together, we believe that these arguments offer a relevant hypothesis and a line of investigation that we will follow in our future studies of this protease.
- “Lastly, can the authors speculate about clinical situations in which reduced Pcsk6 expression may play a role. For example, age or aneurym formation?”
Response 8)
As previously mentioned, our recent studies have shown that PCSK6 expression levels have an important role for SMC activation upon vascular injury and during vascular healing, such as intimal hyperplasia formation, as well as in atherosclerotic plaque formation and rupture [5,11]. Recently, increased expression levels of PCSK6 were shown to be of key importance for cardiac remodeling in cardiomyocytes exposed to hypoxia, both in an experimental in vivo model and in patients with acute myocardial infarction [7]. Genomic variants in the human PCSK6 locus have also been associated with congenital heart disease and aortic dissection [2,3], as mentioned in the introduction (Page 2, line 64-66), as well as blood pressure regulation [6]. Also, high blood pressure and aortic dissection are known risk factors for development of aneurysm, however the presence of PCSK6 in aneurysmatic tissue needs to be further elucidated. We thank the Reviewer for instigating us to expand on this topic and we have now added the following sentences to the discussion part of the manuscript (page 13, line 441-446).
“Human genetic variants in the PCSK6 locus have been linked with increased blood pressure and aortic dissection [3,6], which both are known to be associated with aortic aneurysm formation [12,13]. The risk for development of aortic dissection and aneurysms is increased in patients with genetic diseases resulting in a defective TGFB-signaling [14], where PCSK6 is one of the key proteases involved in the TGFB1 axis [1,15]. However, whether an altered expression of PCSK6 would be associated with aortic aneurysm disease remains to be further investigated.”